



# On the Role of Volcanism in Dansgaard-Oeschger Cycles

Johannes Lohmann[1] and Anders Svensson[1]

[1]Physics of Ice, Climate and Earth, Niels Bohr Institute, University of Copenhagen, Denmark

**Correspondence:** Johannes Lohmann (johannes.lohmann@nbi.ku.dk)

**Abstract.** A significant influence of major volcanic eruptions on regime shifts and long-term climate variability has been suggested previously. But a statistical assessment of this has been hampered by inaccurate synchronization of large volcanic eruptions to changes in past climate. Here, this is achieved by combining a new record of bipolar volcanism from Greenland and Antarctic ice cores with records of abrupt climate change derived from the same ice cores. We show that at >99% confidence bipolar volcanic eruptions occurred more frequently than expected by chance just before the onset of Dansgaard-Oeschger events, the most prominent large-scale abrupt climate changes of the last glacial period. Out of 20 climate change events in the 12-60 ka period, 5 (7) occur within 20 (50) years after a bipolar eruption. Thus, such large eruptions may act as short-term triggers for large-scale abrupt climate change, and may explain part of the variability of Dansgaard-Oeschger cycles.

## 1 Introduction

Volcanic eruptions have been shown to be a major driver of past climate variability (Robock, 2000; Schurer et al., 2013, 2014; Sigl et al., 2015; Swingedouw et al., 2017). Besides the warming trend due to Greenhouse gas emissions, their impact on global mean temperature in historical climate simulations is the only feature that exceeds the ensemble uncertainty in the latest generation of climate models (Tokarska et al., 2020). However, the potential impact of individual very large volcanic eruptions on global climate is not very well constrained due to the very small number of such events to occur since instrumental climate observations began. It is not known whether large volcanic eruptions can drive the evolution of climate on longer time scales, and whether the climate's response can go beyond a linear short-term relaxation after sudden cooling related to stratospheric sulfate aerosols.

Climate variability on the millennial time scale is mostly associated with the Dansgaard-Oeschger (DO) and Heinrich events of the last glacial period. The former consisted of about 30 cycles (Dansgaard et al., 1993) where the cold glacial climate in the Northern Hemisphere (Greenland stadial periods, GS) was interrupted by abrupt warmings of 8-15 K within a few decades (Kindler et al., 2014), followed by prolonged periods of milder but gradually decreasing temperatures (Greenland interstadials, GI). This was often followed by another abrupt transition back to stadial conditions. By synthesizing proxy data from different archives, much progress has been made in unraveling the mechanisms behind these large-scale climate oscillations (Dokken et al., 2013; Lynch-Stieglitz, 2017; Pedro et al., 2018; Sadatzki et al., 2019). At the same time, long simulations with realistic Earth system models became possible, some of which show unforced oscillations of the climate that are very similar to DO



cycles (Klockmann et al., 2018; Vettoretti and Peltier, 2016). Still, a consensus regarding the concrete drivers, if any, that lead to transitions in between stadials and interstadials has not been achieved yet.

One challenge for existing hypotheses concerns the very irregular occurrence times of DO events. While averaging roughly 1500 years, the individual stadials and interstadials that comprise the DO cycles can last anywhere from less than a century up

to ten millennia (Rasmussen et al., 2014). The most realistic model simulations show rather regular self-sustained oscillations of the climate (Vettoretti and Peltier, 2016). However, they currently do not include important factors, such as interactive ice sheets, carbon cycle, and external insolation and volcanic forcing, which might change the nature of the oscillations. This makes it so far difficult to judge whether the simulations are fully consistent with the observed properties of DO cycles: On the one hand, the statistics of the time elapsed before a DO warming transition, as well as the statistics of the durations of the

transitions themselves, are consistent with a purely stochastic driver (Ditlevsen et al., 2005; Lohmann and Ditlevsen, 2019). On the other hand, there is evidence for external influences of insolation, atmospheric $CO_2$ and global ice volume on the lengths of the cycles (Lohmann and Ditlevsen, 2018, 2019), as well as for deterministic features in the data that allow for a prediction of the occurrences of DO events with significant skill (Lohmann, 2019). Nevertheless, such predictions do not perfectly explain the full variability of the occurrence times and leave room for stochastic drivers that influence the timing of event occurrence.

One such driver could be large volcanic eruptions.

A causal relationship between volcanic eruptions and abrupt climate change has been suggested before. The initiation of prominent climate transitions, such as the termination of the last glacial period and the onset of the Younger Dryas cold event have been proposed to be caused by large volcanic eruptions (McConnell et al., 2017; Baldini et al., 2018). While it is difficult to substantiate causality for individual events beyond doubt, a statistical analysis of re-occurring abrupt climate change events

has the potential of establishing a systematic link to external drivers such as volcanic eruptions. This requires records of large volcanic eruptions and climate change events that are as complete as possible, along with a precise age control to tie the eruptions to climatic changes. Those requirements have been challenging for previous studies.

By comparing the largest well-known and absolutely dated volcanic eruptions to climate changes identified in ice cores it was found that within dating uncertainties large Northern Hemisphere (NH) eruptions tend to cluster around the abrupt cooling

phases of DO cycles, whereas large Southern Hemisphere (SH) eruptions might be associated with DO warmings (Baldini et al., 2015). However, the absolute age uncertainties of the layer-counted ice core chronologies as well as those of the radiometrically dated eruptions are typically of the same order of magnitude and grow to more than a millennium during the last glacial period. Similarly, eruptions recorded in an Antarctic ice core and the abrupt cooling events of DO cycles in a Greenland ice core were reported to be clustered closer to another than could be expected by chance (Bay et al., 2004). The same was reported for

abrupt DO warming transitions and volcanic eruptions when allowing for a volcanic lead time of 2 kyr. Still, in this case there were multi-century synchronization uncertainties, limiting the confidence in a direct effect of the eruptions on the climate and making it impossible to judge the temporal order of eruptions and climate change. Additionally, by defining volcanic eruptions from a single ice core this study inevitably included smaller, local eruptions with limited climatic impact.

The present study overcomes both of these issues by using a recently published record of bipolar volcanic eruptions identi-

fied in polar ice cores and the associated synchronized bipolar ice core chronology (Svensson et al., 2020). First, by regarding



volcanic eruptions that led to significant sulfuric deposition at both poles, as seen in Antarctic and Greenland ice cores, all eruptions can be expected to be above a certain threshold in magnitude, and are thus likely to have had large climatic impacts. Compared to the previous study by Bay et al. (2006) with a very sparse record making it difficult to obtain statistical conclusions, the data set employed here contains a much larger number of eruptions. The bipolar matches have furthermore been obtained in a much more reliable way due to parallel layer counting in Greenland and Antarctica in between events. Second, the timing uncertainties between eruptions and climate transitions are greatly reduced by combining this reliable matching of volcanic depositions in Antarctic and Greenland ice cores with a determination of climate transitions from high-resolution isotopic records of the same well-synchronized Greenland ice cores. This allows us to assess the potential occurrence of volcanic eruptions leading up to abrupt climate change with decadal precision.

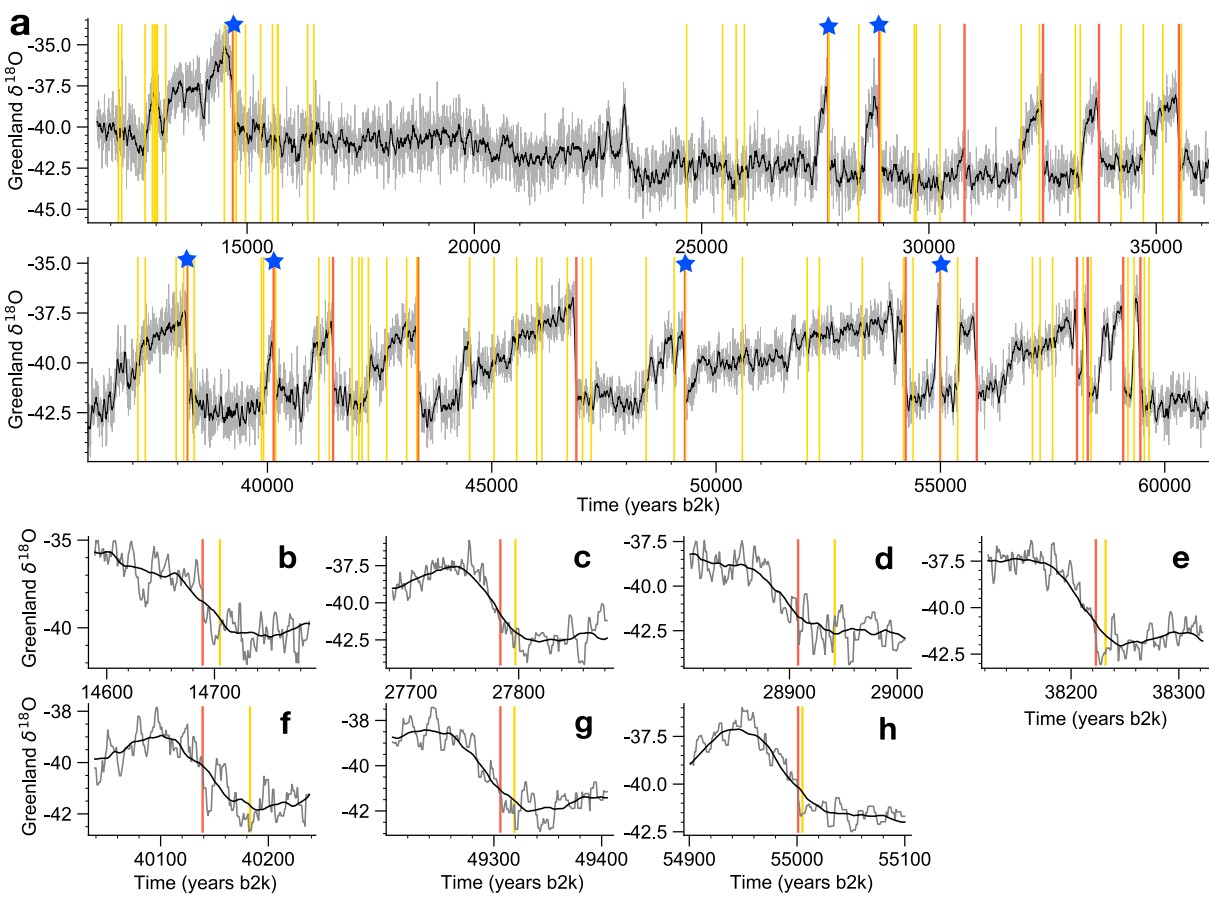

**Figure 1.** Records of abrupt climate change and bipolar volcanism during the last glacial period. **a** High-resolution Greenland $\delta^{18}$O stack (gray, see Sec. 2.1) and 50-year low-pass filtered stack (black), together with the estimated onsets of the DO warmings (red, this work) and the bipolar volcanic eruptions (yellow, Svensson et al. (2020)). For the interval 16.5-24.5 ka no bipolar volcanic eruptions have been identified as the ice cores are difficult to synchronize. The blue stars indicate instances where a volcanic eruption occurs within 50 years prior to the DO warming. The corresponding segments are shown magnified in panels **b-h**.





In the remained of this paper we test the temporal proximity of $N = 82$ bipolar volcanic eruptions to $M = 20$ DO events against a null hypothesis of random and uncorrelated occurrences of volcanic eruptions. Fig. 1 shows our estimates of the DO warming onset times and the bipolar eruption ages from Svensson et al. (2020) together with a stacked Greenland $\delta^{18}$O record. There are a considerable number of DO events that are initiated within a short time after a volcanic eruption (see Fig. 1b-h).

Our main result is that this happens significantly more often than can be expected by chance.

## 2 Methods and Materials

### 2.1 Greenland High-Resolution Isotope Records and Stack

We consider high-resolution $\delta^{18}$O records of four deep Greenland ice cores, as well as a stack derived from these. The records have been measured at different depth resolutions, where each measurement was performed on bulk material of contiguous
depth intervals. The measurements are thus not point samples, but averages over contiguous intervals. We use the $\delta^{18}$O records from the NGRIP ice core (NGRIP Members, 2004; Gkinis et al., 2014), as well as the GRIP (Johnsen et al., 1997), GISP2 (Stuiver and Grootes, 2000) and NEEM (Rasmussen et al., 2013) ice cores. All cores are synchronized to the annual layer-counted Greenland Ice Core Chronology 2005 (GICC05) (Svensson et al., 2006, 2008; Rasmussen et al., 2013; Seierstad et al., 2014). The $\delta^{18}$O records are processed in the following way. The midpoints of the depth intervals are interpolated linearly to
the GICC05 time-depth scale, yielding an unequally spaced time series. Then, this series is oversampled to a 1-year equidistant grid using nearest-neighbor interpolation. Like this, the nature of the measurements as contiguous depth averages and the original measurement values are preserved, and all records are placed on the same equidistant time grid. Finally, we average the four ice cores to obtain a stack with significantly reduced high-frequency noise. We estimate the DO onsets both from the stack and the individual ice cores except GISP2, due to the comparably low resolution of the raw data in the stadials: Starting
at GS-2, the sample resolution decreases from 8 years to around 30 years at GS-15.2. For comparison, the stadial resolutions for the same time periods of NGRIP, GRIP and NEEM are 3 to 6, 4 to 9 and 4 to 8 years, respectively. The time period we consider spans from 11,700 years b2k (years before 2000 AD) to 60,000 years b2k (GS-18). We do not consider the DO events 2.1 and 2.2, as no bipolar matching has been obtained for this interval. As seen in the top panel of Fig. 1, there is effectively a data gap in the volcanic record from 16.5 to 24.5 ka, yielding $\Delta T = 40,300$ years that are investigated.

### 2.2 DO event onset determination

We propose the following method to extract the timing of the DO warming onsets from a high-resolution $\delta^{18}$O record. For each DO cycle, we detect when the data start to deviate significantly upwards from the stadial mean before the abrupt transitions. To this end, we divide the $\delta^{18}$O record into DO cycles comprising one stadial and interstadial defined by the time points given in Rasmussen et al. (2014). Starting at the data point $x_i$ 80 years prior to the expected stadial-interstadial transition midpoint
from Rasmussen et al. (2014), we compute the mean $\mu_{i-1}$ and standard deviation $\sigma_{i-1}$ for all previous stadial data points. The starting point relatively close (80 years prior) to the expected transition is chosen so that no transition onsets are detected





accidentally as a result of the initially large fluctuations in the mean, but the results are not sensitive to the precise value. For very short stadials, we use a starting point at three quarters of the expected stadial duration instead. The data point $x_i$ is then compared with two variable thresholds $u_{i-1} = \mu_{i-1} + \beta_u \sigma_{i-1}$ and $l_{i-1} = \mu_{i-1} + \beta_l \sigma_{i-1}$. We iterate over the data points $x_i$ and keep track of the most recent index $j$ where $x_j$ crosses the lower threshold $x_j > l_{j-1}$, while $x_{j-1} < l_{j-2}$. We stop iterating

when $x_i$ exceeds the upper threshold $u_{j-1}$. The onset is then given by the time point $t_j$.

Since the noise level and the transition amplitude vary between events and ice cores, $\beta_u$ is chosen adaptively. Starting at $\beta_u = 4.5$, we perform the above mentioned routine and check whether the obtained $t_j$ is smaller than a latest reasonable onset time (the time point of the interstadial maximum). If not, we repeat the procedure with reducing $\beta_u$ by 0.1. $\beta_l$ is chosen empirically according to the noise levels of the records, which result from differences in measurement resolution and accumulation. The

estimated onset times of NGRIP ($\beta_l = 0.5$), GRIP ($\beta_l = 0.75$), NEEM ($\beta_l = 0.75$), and the stack ($\beta_l = 1$) are shown in Fig. S2 relative to the first of the four independent determinations. The identified onsets for the isotopic stack are shown together with the isotopic record, the closest bipolar volcanic eruption preceding the onset, as well as the threshold $l$ in Fig. S1.

In the following, we assess the reliability of the onset estimates from the stack and obtain an estimate for the temporal uncertainty. The onsets are very consistent for the events 3, 4, 5.2, 6, 7, 8, 10, 16.1, 16.2, where the range of the four independent

onset estimates is 25 years or less. For the events 1, 5.1, 9, 11, 13, 14, 15.1, 17.1 and 17.2, the stack onset clusters with 2 other ice cores within less than 20 years, while the onset of the remaining ice core is further apart from this cluster and can be considered an outlier. Some outliers may be due to imperfect synchronization, as in the case of events 5.1 and 17.1. Here, the match points used to synchronize NEEM to the GICC05 time scale are spaced several hundred years before and after the DO onset, and thus there is likely an offset in the timing at the onset. In other cases, the shape of the onset in one of the ice cores

is not as abrupt as in the others, or is very noisy, which leads to a late detection with our algorithm. There can also be an early detection when there is a large spike in the high-frequency noise before the onset, as with GI-14 in NEEM. For the remaining events 12 and 15.2, the DO transition appears to consist of two steps in all cores, and due to the high noise level our algorithm is not able to detect the first step in NGRIP, which we argue to be the true onset. To summarize, the onsets derived from the stack should be considered most representative. First, this is because the stack is better suited to define the onsets due to its improved

signal-to-noise ratio. Second, the stack onsets are consistent with the timing estimates from the individual cores. Discarding outliers, we find that the spread of the individual onset estimates (including the estimates from the stack), i.e., the difference of the earliest and latest onset estimate, is 12.9 years on average. This can serve as a good estimate of the uncertainty of the onsets determined from the isotopic stack. Thus, it is possible to assess decadal scale timing differences to volcanic eruptions.

## 2.3 Occurrence statistics of bipolar volcanoes

To test whether DO events preferentially occur shortly after volcanic eruptions, we need to estimate their occurrence rate over time and check the validity of our null hypothesis. In the time interval from 11.7 - 60 ka minus the data gap from 16.5 - 24.5 ka, there are 20641 years of stadial condition with 42 volcanic eruptions and 19659 years of interstadial condition with 40 volcanic eruptions. This yields remarkably similar occurrence rates of bipolar eruptions of $\lambda = 2.0348$ eruptions per kyr for stadials and $\lambda = 2.0347$ for interstadials. Despite the higher noise levels of the sulfur records in stadials compared to interstadials, we





cannot find evidence for a systematic undercounting of eruptions during the stadials. In comparison to the more than 80 bipolar eruptions obtained for the last 2500 years (Sigl et al., 2015), the eruptions considered here are much more sparse. However, this is expected because of the layer thinning of ice cores with depth and due to the much higher background levels of the impurity signals during the last glacial (Mayewski et al., 1997; Schüpbach et al., 2018). Therefore, the sulfate records of the recent

Holocene allow for the detection of much smaller bipolar eruptions. We thus only expect eruptions of very large magnitude to be present in our data set. Five eruptions during the last 2500 years were found to be larger than the Tambora eruption from 1815 in terms of their bipolar sulfuric deposition (Sigl et al., 2015). This corresponds very well to our occurrence rate of two large eruptions per kyr during the glacial.

In the following, we test whether the data are consistent with a stationary Poisson process. First, we need to check whether the

distribution of waiting times $t$ in between eruptions are consistent with an exponential distribution with cumulative probability $P_\lambda(T \geq t) = 1 - e^{-\lambda t}$. This is indeed the case, as seen by an Anderson-Darling (Kolmogorov-Smirnov) test with $p = 0.60$ ($p = 0.66$). The corresponding empirical distribution function along with $P_\lambda(T \geq t)$ using $\lambda = 2.0348$ is shown in Fig. S3a. Second, we confirm that the memoryless property of the waiting times $t$ holds. This is achieved by a two-tailed bootstrap hypothesis test on the Spearman correlation of consecutive waiting times, yielding $p = 0.871$ for the data correlation of $r_S = 0.019$. Finally,

we test the assumption of a constant rate $\lambda$ over time by dividing the volcanic record into short contiguous segments of $\Delta T$ years and testing whether the number $n$ of eruptions in each of them is consistent with a Poisson process at fixed $\lambda = 2.0348$. This is done by calculating the cumulative Poisson distribution function

$$P(N(\Delta T) \leq n) = \sum_{i=0}^{n} \frac{(\lambda \Delta T)^n}{n!} e^{-\lambda \Delta T}. \qquad (1)$$

Choosing $\Delta T = 2$ kyr, except for slightly shorter values at the margins of the investigated time intervals, we find that 2 out

of 21 segments lie outside of the 90% confidence region marked by $P = 0.05$ and $P = 0.95$ (for full results see Fig. S3b). While these 2 segments occur in the youngest part of the record and thus could reflect the higher chances of detecting bipolar volcanic signals due to better signal preservation, we cannot distinguish them from false positives. At 90% confidence we expect 2.1 false positives due to the testing of 21 independent hypotheses. Thus, even with the 2 segments in question, the data do not provide evidence to conclude that the rate of volcano occurrence changed significantly over time. As a result, the data

is consistent with a stationary Poisson process.

## 3  Results

### 3.1  Time lags of DO event onsets to preceding volcanic eruptions

We estimated the precise times of the DO warming onsets from individual high-resolution Greenland $\delta^{18}$O ice core records, as well as from a stack derived from these. Due to the improved signal-to-noise ratio, the stacked record allows for the most

precise onset determination, with an average uncertainty of the onset timings of roughly 13 years (Sec. 2.2). Thus, we will focus on discussing the results derived from the $\delta^{18}$O stack. The time lags of the $M = 20$ DO warming onsets to the closest



preceding bipolar volcanic eruptions are shown in Fig. 2. For the following 7 interstadials the onset occurs within 50 years after a bipolar volcanic eruption: GI-1, GI-3, GI-4, GI-8, GI-9, GI-13 and GI-15.1. If we instead look at a tolerance of 20 years, we find 5 events: GI-1, GI-3, GI-8, GI-13 and GI-15.1.

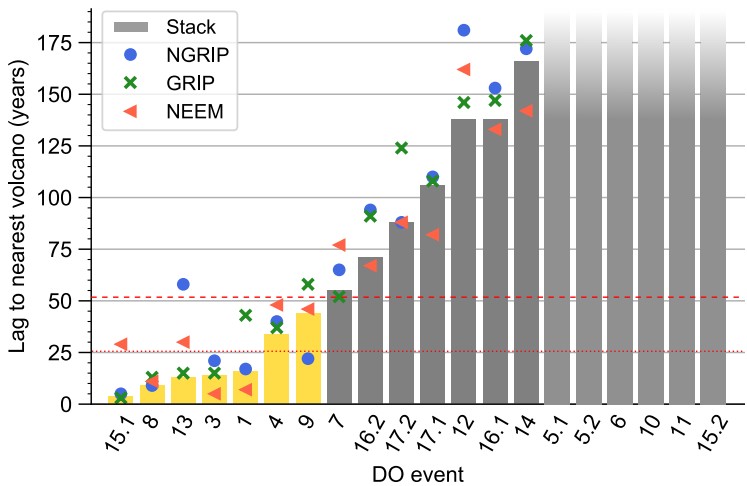

**Figure 2.** Time lag of the DO onsets to the nearest bipolar volcanic eruption. Bars and symbols indicate results for the isotope stack and individual ice cores, respectively. The yellow bars indicate events where the onset estimate in the stack occurs within 50 years after the nearest eruption. The red dotted and dashed lines mark the 5- and 10-percentiles of the time lags for a single event under the null hypothesis, respectively.

## 3.2 Comparison of data to null hypothesis of randomly occurring volcanic eruptions

5   Are the above-mentioned numbers of close-by events likely to be observed by chance in the case that bipolar volcanic eruptions occurred completely randomly and uncorrelated to the occurrence of DO events? To answer this, we test the null hypothesis that volcanic eruptions occur as a stationary Poisson process independently of DO events, i.e., they do not preferentially occur shortly before DO onsets. The description of volcanic eruptions as stationary Poisson process means that they happen at a constant rate $\lambda = N \cdot (\Delta T)^{-1}$ over time and that subsequent eruptions are not correlated (De la Cruz-Reyna, 1991).

10  Our eruption data with $N = 82$ and $\Delta T = 40.3$ kyr are consistent with this assumption, yielding $\lambda = 2.03$ eruptions per kyr (Sec. 2.3). Note that this low occurrence rate implies very large eruptions by historic standards. Only eruptions that represent significant spikes in both Antarctic and Greenland sulfuric records, exceeding the glacial background level of many smaller or local eruptions, are part of the data set. Comparing our rate of eruptions to the record of bipolar eruption of the recent 2500 years (Sigl et al., 2015), the eruptions considered here are of the magnitude of the Tambora eruption or larger.

15  Under the null hypothesis, the expected value for the time lag of an independently occurring DO event to the closest preceding volcano is $\lambda^{-1} = 491$ years. The quantiles for the time lag are $-\lambda^{-1} \ln(1 - p)$. Thus, there is a 5% (10%) chance of observing a time lag smaller or equal to 25.2 (51.8) years. The horizontal lines in Fig. 2 show that in both cases there is a





considerable number of events with smaller time lag. To show that this is indeed significant, we say in the following that there is an event match, if a volcanic eruption occurs within a time lag window of $\tau$ years prior to the DO onset, which we call the tolerance. The number of event matches in the data for tolerance $\tau$ shall be compared to the number of times one or more volcanic events would be found in a time window of $\tau$ years by chance when randomly sampling $M$ windows of a Poisson

process with rate $\lambda$. In this case the number of events $n$ occurring during $\tau$ years is given by the Poisson distribution. The probability to observe one or more events within $\tau$ years is

$$P(n(\tau) \geq 1) = 1 - e^{-\lambda\tau}. \tag{2}$$

Randomly choosing $M$ independent windows and observing whether they contain any events is equivalent to a sequence of Bernoulli trials with success probability $P(n(\tau) \geq 1)$. Thus, the probability of finding one or more events in $k$ out of $M$

windows is given by the Binomial distribution:

$$P_\tau(k) = \frac{M!}{(M-k)!k!}(1 - e^{-\lambda\tau})^k e^{-\lambda\tau(M-k)}. \tag{3}$$

Finally, the probability of finding at least $k$ out of $M$ windows containing one or more events represents the $p$-value for our null hypothesis. This is the cumulative probability of the previous expression

$$p \equiv P_\tau(K \geq k) = 1 - \sum_{l=0}^{k-1} \frac{M!}{(M-l)!l!}(1 - e^{-\lambda\tau})^l e^{-\lambda\tau(M-l)}. \tag{4}$$

For the $k = 7$ matches we find at $\tau = 50$ years, this yields a probability of $p = 0.002$. Thus, we can reject the null hypothesis at a confidence level of 99%. For the $k = 5$ matches found at $\tau = 20$ years we obtain $p = 0.0009$. The expected value of matches is $E[k] = M(1 - e^{-\lambda\tau})$, yielding only 1.9 (0.8) events preceded by one or more eruptions within 50 (20) years under the null hypothesis.

### 3.3   Robustness of results to varying choices of maximum time lag

Our confidence in rejecting the null hypothesis depends on the tolerance $\tau$. To see whether the results are robust, we consider the binomial probability for a plausible range of tolerances. Tolerances lower than 10-15 years should not be considered because they are of the same order as the estimated uncertainty in our DO onset timings. Furthermore, at these tolerances event matches in the data become too rare to be a reliable estimate of the actual probability of co-occurrences of DO events and volcanoes. For tolerances of a century or more, a direct climatic influence of even large volcanic eruptions becomes less likely. Figure 3a

shows the number of matches as a function of $\tau$. The gray line indicates the expected value of matches $E[k] = M(1 - e^{-\lambda\tau})$ under the null hypothesis. The gray (yellow) shading indicates the corresponding 90% (95%) confidence bands. For tolerances larger than 13 years, the data lie consistently above the 95% confidence band, and thus are unlikely to occur under the null hypothesis. The precise probabilities are given in Fig. 3b, where we see that the data lie below $p = 0.01$ for all tolerances larger than 13 years. The confidence at which we can reject the null hypothesis depends on the estimate of the occurrence

rate of eruptions. The shading in Fig. 3b shows the range of probabilities obtained when considering rates $\lambda = N \cdot (\Delta T)^{-1}$ by



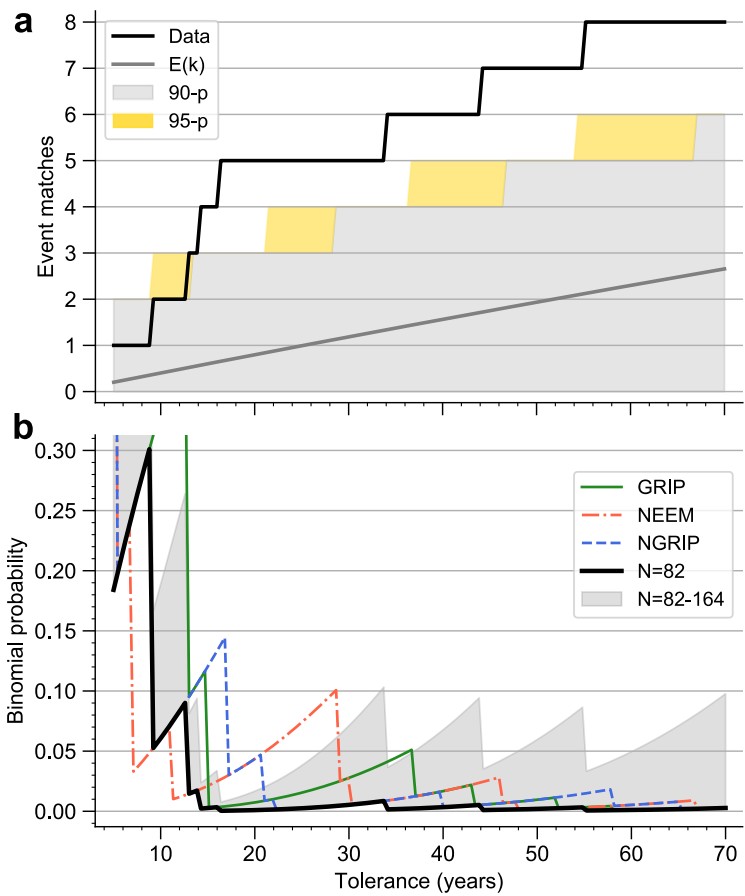

**Figure 3.** Probabilities of the observed number of DO events occurring shortly after volcanic eruptions under the null hypothesis. **a** Number of event matches for the stack onsets (Data) as a function of the tolerance, compared to the expected value $E[k]$ as well as 90% (90-p) and 95% (95-p) confidence bands of the null hypothesis. Here, the upper limit of the bands correspond to the smallest number of matches that have a probability of strictly less than 10% (5%) according to Eq. 4. **b** Probability to observe at least as many event matches as in the data under the null hypothesis. The thick black line shows the results for the stack onsets and the Poisson null hypothesis using $\lambda = N/\Delta T$ with $N = 82$, being the number of large eruptions in the investigated time interval. The gray shading gives indicates the indicates the range of probabilities when doubling $N$ from $N = 82$ to $N = 164$. Results for the onsets from individual cores with $N = 82$ are also shown.

doubling $N$ from 82 to 164. Even then we consistently find $p \leq 0.1$, allowing us to reject the null hypothesis at 90% confidence. Thus, our results are robust to a potential under-counting of large bipolar eruptions where half of the eruptions not occurring close to a DO event had not been detected. Furthermore, the results are also significant when considering the onset timings of the individual ice cores as can be seen in Fig. 3b.



### 3.4 Volcanic influence on terminations of DO interstadials

Our results might surprise the reader since it is more intuitive for volcanic eruptions to induce the abrupt cooling transitions
of DO cycles due to the radiative aerosol cooling. We performed the same analysis on the cooling transitions that mark the
terminations of interstadials and do not find a significant clustering of eruptions leading up to the coolings (see Fig. S4). We
note, however, that the cooling transitions in the isotopic records are much less well-defined. Thus, their timing estimates,
determined here by a different method presented in Lohmann and Ditlevsen (2019), have a much larger uncertainty. As a
result, we might have missed individual event matches, such as the volcanic eruption occurring close to the termination of
GI-16.2. The short interstadials GI-16.2 and GI-17.2 (where we also find a volcanic eruption close to the termination) are
outliers (Lohmann and Ditlevsen, 2019). Thus, a volcanic influence to end them prematurely might be plausible. Still, despite
the timing uncertainties it seems evident that such volcanic influences before DO coolings did not occur more frequently than
would be expected by chance.

### 4 Discussion and Conclusions

Here we present evidence that initiations of DO events within a short time after large volcanic eruptions happened more fre-
quently than can be expected by chance. Whereas recent research showed that DO event onsets are predictable in principle
(Lohmann, 2019), there could be additional factors that influenced the timing of some abrupt warmings, such as the perturba-
tions invoked by the volcanic eruptions identified here. These could be interpreted as short-term, stochastic triggers influencing
the timing of DO onsets, which, in the absence of such triggers, might eventually occur regardless due to processes that happen
on longer time scales. Indeed, for 3 out of 4 events where we find an eruption within 20 years prior to the onset, it occurred
earlier than predicted in Lohmann (2019) (GI-1 was not part of that study).

Using a new record of bipolar volcanic eruptions in conjunction with multiple high-resolution Greenland ice core records, it
is for the first time possible to investigate this problem with the required temporal precision, so that a direct climatic influence
of eruptions on the abrupt climate transitions is plausible without having to invoke centuries-long climate feedbacks. When
determining significance of the results, our method uses tolerance windows and is thus not sensitive to errors in the timing of the
onsets on a sub-decadal time scale. This includes small potential offsets in the initiations of DO warmings when estimated from
other proxies (Erhardt et al., 2019). Since only eruptions that happen within the tolerance enter the calculation, the results are
not affected by any unidentified eruptions in the stadials. This allowed us to show that the results are robust under considerable
uncertainty in the total number of eruptions.

A limitation of the study is the unknown eruption magnitudes. Even though the eruptions are known to have emitted large
amounts of sulfate into the stratosphere due to their bipolar signature, there are certainly differences in the magnitudes. Further
studies quantifying the sulfate deposition in the ice cores are underway, which will allow a more precise assessment of their link
to abrupt climate change. An initial analysis shows that the eruptions occurring shortly before DO events are a representative
sample of the ensemble of all bipolar eruptions in terms of their magnitude. Knowledge of the relative magnitudes will also
allow a more nuanced treatment of the statistical process underlying the eruptions. While our data suggests that the volcanic



eruptions themselves are consistent with a stationary, random process and thus can be considered unpredictable, more research

is needed to assess the uniformity of volcanism over the last glacial period, which has been challenged by other studies (Huybers and Langmuir, 2009; Brown et al., 2014).

Contrary to previous studies (Bay et al., 2004, 2006; Baldini et al., 2015), we do not find evidence for a causal relation of eruptions preceding abrupt cooling events, but preceding abrupt NH warming instead. This might seem counter intuitive since large eruptions are associated to first order with a global cooling response. However, the actual climate response can

be more complicated, depending on both the site of the eruption as well as the season (Robock, 2000). The radiative aerosol cooling is typically not uniformly distributed over the globe, and for eruptions occurring at higher latitudes, the response is hemispherically asymmetric (Pausata et al., 2015). In turn, tropical eruptions lead to stronger cooling at the tropics, which results in altered equator-pole temperature gradients (Pausata et al., 2020). These asymmetries lead to changes of ocean and atmosphere circulation, which can impact the global climate in different ways. Via northward (southward) shifts of the inter-

tropical convergence zone, this asymmetric cooling of SH (NH) volcanic eruptions was purported to be able to initiate abrupt DO warming (cooling) in the NH (Baldini et al., 2015), when amplified by further feedbacks likely associated with sea-ice extent in the North Atlantic (NA), which is known to vary strongly between stadial and interstadial periods (Hoff et al., 2016). Since the locations of the bipolar eruptions investigated here are unknown at the current stage, we cannot make a similar distinction. The eruption latitudes will be constrained in a future study, allowing a more detailed investigation of potential

mechanisms.

DO warming events are generally associated with an abrupt strengthening of the Atlantic Meridional Overturning Circulation (AMOC), and there have been several modeling studies that show a direct AMOC strengthening as a response to NH high-latitude (Pausata et al., 2015) as well as tropical eruptions (Stenchikov et al., 2009; Swingedouw et al., 2015). If the climate system is in a bi-stable state, related for instance to the positive salt-advection feedback of the AMOC, and close to a tipping

point, relatively small, direct perturbations of the AMOC could lead to a transition from stadial to interstadial conditions. Alternatively if the system displays self-sustained relaxation oscillations of the AMOC, such perturbations could lead to a premature advance of the cycle. A mechanism that has been proposed for an abrupt strengthening of the AMOC is an instability of the ocean stratification (Dokken et al., 2013), which may be initiated by abrupt changes in NA sea ice cover (Vettoretti and Peltier, 2016). Sea ice cover is in turn sensitive to wind stress. In climate model simulations, changes in zonal wind have been

shown to alter the NA sea ice cover such that abrupt strengthening of the AMOC and NA warming is initiated (Zhang et al., 2014). A similar situation might occur after a large volcanic eruption, where the jet stream and zonal winds respond to altered meridional temperature gradients (Robock, 2000; Pausata et al., 2020). More firm conclusions regarding the mechanism can only be achieved as additional data or model simulations become available. Nevertheless, the statistical link in between large volcanic eruptions and abrupt climate change shown here enhances our understanding of the causes of abrupt climate change,

as well as of potential future impacts of volcanic super-eruptions.



*Data availability.* The volcanic record is available in the supplementary material of (Svensson et al., 2020). The high-resolution NGRIP oxygen isotope record is available at http://iceandclimate.nbi.ku.dk/data/NGRIP_d18O_and_dust_5cm.xls. The GISP2 record is available at http://depts.washington.edu/qil/datasets/gisp2_main.html. The NEEM high-resolution oxygen isotope record is currently being published on PANGAEA by Gkinis et al. of the ice and climate group at Copenhagen University, and will be available shortly. The GRIP record is
5   available upon request from the corresponding author.

*Competing interests.* The authors declare no competing interests.

*Acknowledgements.* This project is TiPES contribution #68. The project has received funding from the European Union's Horizon 2020 research and innovation programme under grant agreement No. 820970.





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
