# Peer review of "On the Role of Volcanism in Dansgaard-Oeschger Cycles"

_Climate of the Past, 2020_

## Referee Comment (RC1) · Anonymous Referee #1 · 12 Feb 2021

This is a potentially intriguing paper. It takes an existing database of bipolar volcanic signals (from Svensson et al 2020), and looks at the timing of eruptions relative to the onset of Dansgaard-Oeschger warmings (GI onsets) in Greenland. The main result is that eruptions in the database occur within 20 years of a DO onset more often than can be expected by chance. As a result it suggests that eruptions can help to push the AMOC (assumed to be the agent responsible for the DO event) over the edge.

If correct this is exciting. By no means all DO events have such a volcano near their onset, and large eruptions take place during glacials without causing a DO warming, so the conclusion would be reasonably stated as showing that eruptions are neither necessary nor sufficient for a DO event but they can precipitate the start of an event that would have happened eventually anyway. The authors don't really have a mechanism for why the volcano should cause such an acceleration of the event but just putting the idea out there would no doubt encourage new research and ideas.

But the issue is whether they have really shown what they claim. The statistics they use need to be looked at by a statistician but I am convinced that they are correct that the events logged in the Svensson et al (2020) paper do occur close to a DO warming more often than can be explained by chance. However, are those events a representative set of bipolar eruptions? I have my doubts, and I think the authors need to consider and (find ways to reject) an alternative.

Firstly the database contains only 81 events in 40 kyr of data. As the authors point out themselves this return rate of 1 every 500 years, is far below what is observed in the top 2400 years. Sigl et al (2015) found 81 bipolar eruptions in 2400 years (1 every 30 years) of which they classed 40 (1 in 60 years) as "large" (meaning a bit bigger than Pinatubo and a bit less than half the strength of Tambora). So we can already see that it is not surprising at all that there is a bipolar eruption within 20 (or 50) years of many DO warmings – if there is something special about the smaller subset considered by Svensson it needs to be that the chosen eruptions are large – the 1 in 500 year size of the 1458 or 1257 eruptions.

The present paper argues that they see fewer eruptions than Sigl because only very large eruptions exceed the background of the glacial sulfate and the lower resolution caused by thinning. However this seems unlikely to be the issue: while other chemistry changes a lot, the nssS background in the glacial (McConnell data online) is not very much higher than that of the Holocene. Volcanic eruption peaks are easily picked out even at sites with much lower resolution such as Dome C, and the layer thickness at WAIS Divide is still about 2 cm/year even at 50 ka age. Furthermore the peaks that occur near to terminations are by no means large ones. I didn't investigate all of them, and it's hard work to assess the peaks because the 2020 paper gives no scale for the y-axis but if we take the first 3 DO warmings where a bipolar volcano is identified within 20 years:

For the Bolling onset, eruption at 14705 years (GICC05), we can see in Svensson et al (2020) that it's a pretty modest peak in all records. It's barely visible at EDC, about the 6[th] largest peak between 14500 and 15000 years at EDML, while at NGRIP and WDC there are 8 peaks larger than this one in the 2000 years shown in Fig S2A of Svensson et al (2020). In WDC (using the online data) it rises about 50 ppb S (150 ppb sulfate) above the background. I estimated the flux from its width and the accumulation rate, and estimate that it's a modest eruption that would not meet Sigl's definition of a "large" eruption. In any case it is clearly not a 1 in 500 years eruption n any of the records.

For the onset of GI3, eruption at 27797 years, it's again a very modest peak. It's not really visible in any of the electrical records shown in Fig S4A of Svensson et al (2020) and it's a tiny peak in the only

Antarctic S record shown (WDC) – barely above the background (about 30 ppbS above the background), and a factor 10 smaller than several peaks in the 3000 years shown on the figure.

Finally for the onset of GI8, at 38232 years, it's again a modest peak in WDC, and in Greenland cores, the peak is less than half the size of a peak that preceded it by a century.

So, I would contend that the peaks near terminations are not especially large, and this cannot explain why they are the chosen subset of the much more numerous bipolar eruptions that exist. This then raises the suspicion that the reason there are more bipolar eruptions near GI onsets is something different: did the authors of Svensson et al (2020) find it easier to identify bipolar eruptions when they are near to DO onsets, or did they concentrate their efforts on the time periods around DO onsets.

The first issue (easier to identify bipolar eruptions near onsets) is obvious: these are the points where the relative age of Greenland and Antarctic cores are rather constrained by the existing methane matches that occur at DO onsets. This makes it much easier to be confident two eruptions in the two hemispheres that might match are actually at the same date. Hundreds of years before the DO onset, the relative age models are rather unconstrained and the authors of the 2020 paper would have felt less confident in calling two peaks near each other bipolar.

In addition the purpose of the 2020 paper was to look at the timing of events in Greenland and Antarctica at DO onsets. As a result, by their own admission they put a lot more effort into finding tiepoints around the DO warmings, which they did more confidently when they had layer counting. They reported "The bipolar layer counting is not continuous but is **focused on periods of abrupt climate variability** or high volcanic activity". They also noted "In addition to the published volcanic match points made for Antarctica, some 25 additional Antarctic match points have been identified in the present study to strengthen the synchronization **in the neighbourhood of Greenland abrupt climate change events**."

There is nothing surprising or bad about this, but it's just a fact that Svensson et al (2020) wanted matchpoints near the DO warmings and so they put more work into those areas, which are also the areas where it's easier to pin the relative dates down. As an example if I look at the section near the start of GI3 (27800 to 28500) I can see by eye at least 2 other potential bipolar matches of peaks in both WDC and NGRIP of similar size to the one identified at the onset, but where (correctly) the relative uncertainty in dating precluded marking them with certainty.

In conclusion, I believe that this is a topic worth studying, and I agree that the peaks in the Svensson (2020) compilation occur close to GI onsets more often than would be expected. However I am doubtful, using the information given in that paper, that the 81 bipolar peaks identified are an unbiased sample in time. I am not entirely sure how the authors can overcome my concerns. Probably the only way would be to run an algorithm that seeks possible bipolar matches within the prior (ie based only on methane) relative age uncertainty, so that an unbiased sample of possible bipolar peaks can be compiled and compared to the timing of GI onsets.

The authors tell us they are working on looking at the magnitude of the eruptions they identified and perhaps this will help them. It remains a fact that (based on Sigl's work) it is more likely than not that a modest bipolar eruption will occur by chance every 30 years and a "large" one every 60 years, so unless the authors can show that these ones near GI onsets are the rare "very large" ones, then they have not discovered anything very interesting.

In addition to this long and major comment above, I have some less important points:

Page 2, line 35 "by regarding" is strange wording, probably "by concentrating on"

Page 4  line 16 "Like this" should be "Thus" or "In this way"

Page 5, lines 1-5. I have now read this several times but I still can't follow what you did. I get that you are seeking a point that exceeds the noise of the glacial data, but in detail the method is hard to understand. Did you start with the mean of the points within 80 years of the onset or before 80 years (and if so how far back?). For short stadials, is the set point 75% of the way back to the next GI or 75% since the last GI? Where is $t_j$ – at the lower threshold or the upper threshold. I think you would need a section in the supplement that explains this clearly, with a diagram, and also explains why you chose this method over previous methods such as used by Rasmussen, or by those using Rampfit type methods. Having said that I accept that the points you have deduced are reasonable, so I am not objecting to the method but simply couldn't follow it from the text provided.

Page 7, lines 10-15 essentially repeats what you already said at page 6 lines 5-8: needs redrafting.

---

## Referee Comment (RC2) · Reik Donner (Referee) · 15 Mar 2021

The authors of this interesting discussion paper study the temporal co-occurrence of Dansgaard-Oeschger (DO) event related rapid glacial climate changes inferred from a suite of published Greenland ice cores with synchronous sulfur spikes in bipolar high-latitude ice cores indicative of strong volcanic eruptions. They report detectable volcanic signatures within very few decades prior to the onset of DO events to occur more frequently than expected by chance. This finding may be indicative of strong volcanic activity superimposed to a complex and intrinsically bistable dynamics being able to provide the essential "momentum" for the climate system close to a transition point becoming forced to tip to the other state. This is an interesting observation –

generally consistent with our simplistic mechanistic understanding of the behavior of stochastic bistable systems – that clearly deserves being reported here and further studied in future work. It is important to note that the analysis presented here does not strictly imply evidence of causality between episodes of strong volcanic activity and the timing of DO events (yet provides indications that there might be some kind of statistical linkage), while a more detailed mechanistic explanation of such a link remains still somewhat vague. This is however not to be understood as a criticism to the present study, which raises a valuable point requiring future exploration.

Besides addressing a previously largely unexplored aspect of glacial climate variability, the present work is innovative in bringing together very recent paleoclimate datasets (with full awareness of their intrinsic uncertainties) with a simple yet appropriate statistical/stochastic analysis methodology. Potential issues regarding the employed datasets and their possible limitations have already been discussed in great detail by another reviewer, so that I do not want to further elaborate on this aspect here in detail.

One point of concern could be the fact that a large time interval (24.5-16.5 ka b2k) has not been analyzed, not because of an absence of DO events, but due to a lack of synchronicity of bipolar ice cores required for the identification of episodes of very strong volcanism. (So we do not have an absence of volcanic activity during that time period, but just an absence of synchronizable events, as far as I understand the present work.) This implies that the available record of bipolar sulfur anomalies in ice cores has to be cut at specific points that largely reflect the author's choice. Since the affected time interval includes only a single DO event (GI-2), this might not affect the statistic much at first glance. However, in the end this depends markedly on the specific formulation of the statistical hypothesis addressed, which should be further clarified in a revised version of this interesting manuscript.

In a nutshell, the authors use an analysis methodology that is largely equivalent to event coincidence analysis (ECA) introduced by Donges et al. (2011, 2016). In what follows, I will adopt their notion, being fully aware of the corresponding limitations. ECA

is based on "counting" the number of co-occurrences between two types of events, where co-occurrence is defined by introducing two parameters. (a) A time shift parameter (allowing the quantification of delayed responses, which is essentially assumed to be zero by the authors of the present study, a setting that may be questioned according to substantial inertia of the studied type of climate transition essentially happening in the Atlantic ocean). (b) A tolerance window (addressing time uncertainty of events along with possibly distributed response times, which is also considered by the authors of this study). One question to the authors would therefore be if it would not make more sense for their work to also consider a minimum response time instead of only a maximum possible delay (as their free parameter "Tolerance").

Next, ECA can be employed to address two related yet different research questions. (a) The likelihood of an event of type A (DO event) to be closely preceded by at least one (the latter is important, since events may also cluster in time) event of type B (sulfur spike/strong volcanic activity). This is what Donges et al. refer to as "precursor event coincidence rate". (b) The likelihood of an event of type B closely preceding at least one event of type A (in the present context, "at least" would be irrelevant since DO events to not occur in such close succession). Donges et al. termed this "trigger event coincidence rate".

Along with the former distinction, we get to the point where the missing time window around GI-2, as well as the definition of the full time span of observations becomes potentially relevant. When using precursor rates, we only need to fix the timing of known DO events within the admissible period of observations and do not need to take care of its actual span. There is no interest in using sulphur spikes as predictors for approaching DO events, and hence, there are no "false positives". On the other hand, one could also be interested in the reverse research question of the predictability of DO events based on volcanic eruptions. In the present case, the latter question does not make much sense to me scientifically (DO events obviously require a certain preconditioning of the climate system, and volcanic activity acts mainly to determine

the actual timing of the event to happen, according to the research hypothesis of the authors). However, this question would also be partially ill-posed due to the imperfect time coverage of the identified sulfur events and the "subjective" choice of the exact time period considered for such an analysis.

I do not discuss this aspect here in such great detail because I see any obvious error in the analysis presented. However, (i) it might be worth clarifying those points in the manuscript – also in relation to the concerns of the other reviewer - and (ii) it appears to me important to be precise about what kind of research question the presented analysis is able to address, and which not. This clarification also helps motivating why the analysis setting used in the present work is appropriate. If, for example, DO events would not always be separated by much longer time intervals than the tolerance window used for the analysis (or if they were not sufficiently rare), this would render the assumption of a binomial distribution for studying coupling between two presumed Poisson processes an invalid basis for a test statistic. This aspect has also been discussed in detail by Donges et al. (2016).

Regarding the latter assumptions beyond the binomial distribution of event co-occurrences, I am however wondering about another aspect. The authors have well addressed the waiting time distributions between sulfur anomalies and reported convincing evidence for an absence of significant deviations from an exponential distribution, supporting the hypothesis that the volcanic eruption episodes can be well approximated as an uncorrelated temporal point process (Poisson process). However, in order for the employed analysis to make full sense, it would also be required that the DO events follow a Poisson process as well, which is clearly not the case due to the known shape of the waiting time distribution peaking at a 1.5 ka time scale. Strictly speaking, if one of the two type of events violates the assumption of uncorrelated events in such prominent way, the confidence bounds derived from the theoretical reasoning in Section 2.3 may not apply anymore. To clarify this aspect, I recommend a simple numerical check based on random event sequences leaving the waiting time distributions of both

types of events invariant. A possible alternative would be repeating the analysis as performed in the manuscript, but successively shifting the series of volcanic events relative to the DO series and counting how often the empirical co-occurrence frequency between both types of events is reached at time shifts for which one would not expect any statistical link between both to apply.

Other comments:

1. Can strong sulphur anomalies in bipolar ice cores really be attributed to single strong eruptions? Or would it also make sense to check for close successions of events, each of which providing a small "kick" to the system towards the point of instability? Regarding Late Holocene climate variability (e.g., the transition between Medieval Climate Anomaly and Little Ice Age), it had been reported that in addition to solar variability, an enhanced period of volcanic activity might have had a crucial impact for this millennial scale transition to occur (e.g. via bistability of the North Atlantic subpolar gyre, cf. Schleussner et al. 2015). This is of course not strictly related to the problem studied in the present work, but I wonder if one might draw upon some analogies here.

2. It might be important to study more systematically the effect of different magnitude thresholds to sulfur spikes in bipolar ice cores (and, hence, the density of identified volcanic events) on the results of the present study.

3. In terms of stacking different Greenland ice cores, I am wondering about the relevance of interpolation suppressing high-frequency variability. At which time scale is high-frequency variability actually uncorrelated among the different cores, and what would this imply for the rationale of high-frequency variability just being considered as noise? I suppose this is mainly related to the aspect that DO events have a strong (slow) ocean component while being forced here by high-frequency (intra-annual to inter-annual) atmospheric variations carrying the volcanic signal. It might be helpful if the authors could elaborate a bit further on this aspect in Section 2.1.

4. In Section 2.2, the definition of the "stadial mean" used as a benchmark for identifying the DO event onset appears a bit opaque to me. For sure, both stadial mean and variance may depend markedly on the actual reference period taken into account. Related to this, as well as to the previous comment: What do the authors define as "noise level"?

5. I suppose that "nearest eruption" in Fig. 2 etc. always refers to the nearest preceding eruption. This should be clarified in the caption. Is it actually possible to uniquely identify this term, given remaining statistical uncertainty regarding the timing of DO event onset?

6. Can you elaborate a bit more on the plausibility of possible mechanisms of volcanic activity triggering DO events at decadal or even multidecadal lags?

7. In my opinion, the manuscript title might deserve a bit more precision - the manuscript topic is quite a bit narrower than the title suggests.

Technical comments:

- p.4, l.1: "In the remainder..."

- p.4, l.19: "except for GISP 2"

- p.7, l.13: "eruptions"

- p.9, Fig. 3, caption, l.6: remove "gives indicates the" (same also in Suppl. Mat., Fig. S4)

- Suppl. Mat., Fig. S3, caption, last line: "hypotheses"

References used in this review:

J.F. Donges, R.V. Donner, M.H. Trauth, N. Marwan, H.J. Schellnhuber, J. Kurths: Nonlinear detection of paleoclimate-variability transitions possibly related to human evolution. Proceedings of the National Academy of Sciences of the U.S.A., 108(51), 20422-20427 (2011)

J.F. Donges, C.-F. Schleussner, J.F. Siegmund, R.V. Donner: Event coincidence analysis for quantifying statistical interrelationships between event time series - On the role of extreme flood events as possible drivers of epidemics. European Physical Journal - Special Topics, 225(3), 471-487 (2016)

C.-F. Schleussner, D. Divine, J.F. Donges, A. Miettinen, R.V. Donner: Indications for a North Atlantic ocean circulation regime shift at the onset of the Little Ice Age. Climate Dynamics 45, 3623-3633 (2015)